# Molecular Signatures of Aeroallergen Sensitization in Respiratory Allergy: A Comparative Study Across Climate-Matched Populations

**DOI:** 10.3390/ijms26010284

**Published:** 2024-12-31

**Authors:** Ruperto González-Pérez, César Alberto Galván-Calle, Tania Galán, Paloma Poza-Guedes, Inmaculada Sánchez-Machín, Oscar Manuel Enrique-Calderón, Fernando Pineda

**Affiliations:** 1Allergy Department, Hospital Universitario de Canarias, 38320 Tenerife, Spain; pozagdes@hotmail.com (P.P.-G.); zerupean67@gmail.com (I.S.-M.); 2Severe Asthma Unit, Hospital Universitario de Canarias, 38320 Tenerife, Spain; 3Instituto de Investigación Sanitaria de Canarias (IISC), 38320 Tenerife, Spain; 4B&D Salud e.I.R.L., Lima 15300, Peru; cegals8@hotmail.com; 5Inmunotek SL Laboratories, 28000 Madrid, Spain; tgalan@inmunotek.com (T.G.); fpineda@inmunotek.com (F.P.); 6Allergen Immunotherapy Unit, Hospital Universitario de Canarias, 38320 Tenerife, Spain; 7Clínica SANNA el Golf, Lima 15300, Peru; oscarcalderonll@gmail.com

**Keywords:** aerobiology, exposome, allergens, allergic rhinitis, allergic asthma, climate change

## Abstract

Climate change is significantly altering the dynamics of airborne allergens, affecting their seasonality, allergenicity, and geographic distribution, which correlates with increasing rates of allergic diseases. This study investigates aeroallergen sensitization among populations from Tenerife, Spain, and Lima, Peru—two regions with similar climates but distinct socio-economic conditions. Our findings reveal that Spanish individuals, particularly those with asthma, demonstrate higher sensitization levels to a broader range of allergens, especially mites, with 85% of participants reacting to at least one mite allergen. In contrast, Peruvian patients exhibit a narrower spectrum of sensitization. These results highlight the influence of environmental factors, such as pollution and socio-economic disparities, on allergen exposure and immune responses. Moreover, this study underscores the necessity for region-specific diagnostic and therapeutic strategies to effectively address these variations. By elucidating the intricate relationship between climate change, environmental factors, and allergen sensitization, this research offers insights into respiratory allergic conditions, advocating for tailored interventions to mitigate their impact across diverse populations.

## 1. Introduction

Recognizing the impact of the external exposome on allergy development is becoming increasingly urgent. Environmental exposures can be classified into three interconnected domains: the general external environment; the specific external environment; and the internal, host-dependent environment [1]. The general external environment encompasses factors such as climate, biodiversity, and the urban, social, and economic conditions that shape human interactions with the natural world [2,3]. In contrast, the specific external environment involves direct exposures, including allergens, microbes, diet, tobacco, pollutants, and other toxic substances that directly impact human health. In addition, the internal environment consists of host-dependent physiological factors—such as metabolism, inflammation, and oxidative stress—that determine how individuals respond to these external exposures [4,5,6]. Together, these three domains form the concept of the “meta-exposome,” which reflects the dynamic interplay between environmental exposures and human health [7,8].

Allergen exposure is a primary factor in allergic sensitization, a critical process in the development of conditions like asthma, allergic rhinitis, and eczema [9,10,11,12]. Airborne allergens, particularly pollen and dust mites, are the most significant risk factors for respiratory allergic diseases. Moreover, demographic factors such as age, sex, and geographical location also influence patterns of allergen sensitization [13,14,15,16].

Climate change is a major threat to humanity, driven by greenhouse gases that pose significant challenges to human health and healthcare systems, potentially reversing decades of medical progress [17]. One notable health impact is the rise in allergic respiratory diseases, fueled by increasing atmospheric carbon dioxide and higher temperatures. These factors intensify the concentration and allergenicity of airborne particles like pollen and fungal spores, resulting in more severe symptoms [18]. Additionally, the biodiversity hypothesis suggests that exposure to natural environments enriches the human microbiome and promotes immune balance, offering protection against allergies and inflammatory disorders. However, the loss of these immunoprotective factors in rapidly urbanizing areas experiencing biodiversity loss is an emerging health risk [19,20].

Understanding the external exposome’s role in allergy development is critical. Advances in Precision Allergy Molecular Diagnosis (PAMD@) now allow for more precise identification of individual IgE reactivity profiles, leading to more accurate diagnoses and personalized treatments [21,22,23,24]. The aim of this study was to investigate regional patterns of allergic sensitization in two distant regions—Tenerife, Spain, and Lima, Peru—which, despite sharing similar climatic conditions, differ significantly in urbanization and socio-economic status. Evaluating how these factors influence allergen exposure in different geographic areas is essential for developing targeted diagnostic and therapeutic tools.

## 2. Results

### 2.1. Demographic Features of Investigated Patients

All 181 subjects who met the ARIA or GINA criteria for allergic rhinitis (AR) or asthma (A) [25,26] tested positive for one or more aeroallergens in a skin prick test (SPT). Most participants were female (67%) with a median age of 29.7 years (range: 4–75) (Table 1). 

None of the patients had received allergen immunotherapy or biologic treatments before or during the study inclusion. Atopic comorbidities were present, with food allergies (to seafood, nuts, eggs, and/or milk) affecting 9.42% (18 patients) and drug allergies and/or hypersensitivities—to beta-lactam antibiotics and/or nonsteroidal anti-inflammatory drugs—in 6.62% (12 patients). Additionally, a family history of atopy was reported by 74.25% of the patients.

### 2.2. Prevalence, sIgE Reactivity, and Individual Molecular Profile According to Atopic Disease

The sensitization to aeroallergen extracts through SPT and its prevalence in 181 patients who met the inclusion criteria are summarized in Table 2.

Overall, 162 out of 181 patients (89.5%) were sIgE-positive (≥0.35 kUA/L) to one or more of the 52 individual molecular aeroallergens included in the multiplex array (Figure 1).

Despite having a positive SPT for at least one of the investigated aeroallergens, 19 patients (10.4%)—comprising 12 with AR (11 from Peru and 1 from Spain) and 7 with AA (6 from Peru and 1 from Spain)—did not display specific IgE (sIgE) levels above 0.35 kUA/L for any of the allergens tested in the multiplex array.

### 2.3. Mites

Mites were identified as the most prevalent source of sensitizing airborne allergens in both populations, regardless of the subjects’ underlying atopic condition. Sensitization to one or more of the 17 investigated mite molecular allergens was observed in 154 (85%) subjects. However, 27 patients—19 with AR (18 from Peru, 1 from Spain) and 8 with AA (7 from Peru, 1 from Spain)—were not sensitized to any of the mite allergens tested. Our findings confirmed Der p 1, Der p 2, Der f 2, and Der p 23 as major allergens (prevalence above 50%). Mid-tier allergens, with a prevalence between 20% and 50%, included Der f 1, Der p 5, Der p 7, Der p 21, Blo t 5, Blo t 21, Gly d 2, Lep d 2, and Tyr p 2. Minor allergens, with a prevalence below 20%, were Der p 10, Der p 11, Der p 20, and Blo t 10.

The overall proportion of subjects with sIgE to group 1 allergens—Der p 1 and Der f 1—was 53.1%, which was lower than to group 2 allergens—Der p 2 and Der f 2—at 63.7%, and to Der p 23 at 61.6%. Among the individual molecular allergens, Der p 2 (63.8%) was the most frequently identified with sIgE ≥ 0.35 kUA/L, followed by Der f 2 (63.6%), Der p 23 (61.6%), Der p 1 (59.6%), and Der f 1 (47.5%). Among the subjects sensitized to group 1 allergens (53.1%), 79.8% were positive for both Der f 1 and Der p 1 (21 subjects were exclusively sensitized to Der p 1 and 1 subject to Der f 1). Similarly, 99.1% of subjects sensitized to group 2 allergens had sIgE to both Der f 2 and Der p 2 (with only one subject having sIgE exclusively to Der f 2).

A majority of subjects (57%) demonstrated recognition of 9 to 16 individual mite allergens (any), with the predominant IgE profile—consisting of 11 molecules—covering all five major allergens (Table 3). Isolated sensitizations to individual mite allergens were infrequent, occurring in only 2.76% of cases, with exclusive sensitization to Der p 23 observed in a single patient (0.5%) out of 181.

#### 2.3.1. sIgE Reactivity Profiles and Basal Respiratory Allergic Diseases by Geographic Location

The overall frequency of molecular sensitization was quantitatively higher in 13 out of 17 (73.47%) molecules—Der p 1, Der p 2, Der p 5, Der p 7, Der p 11, Der p 20, Der p 21, Der p 23, Der f 1, Der f 2, Blo t 5, Lep d 2, and Tyr p 2—in patients with AA, in contrast to those with AR, who were more frequently sensitized to Der p 10, Blo t 10, Blo t 21, and Gly d 2. In addition to the high number of sIgE mite responses, significant differences in prevalence and quantification titers were observed between the basal atopic diseases and their corresponding mite molecular allergens in patients with AR and those with AA.

Individuals with AR in the Spanish cohort showed a higher overall frequency of sensitization to all 17 mite molecules compared to those in the Peruvian cohort. Interestingly, for those with AA, only one molecule (Blo t 21) showed higher sensitization in the Peruvian population compared to the Spanish population. Spanish asthmatic patients demonstrated increased sensitization to most mite allergens compared to those with AR, except for Blo t 21, Gly d 2, and Tyr p 2. Similarly, Peruvian asthmatic subjects showed a higher frequency of sensitization to most individual allergens compared to those with AR, with the exceptions of Der p 10, Blo t 10, and Tyr p 2. Spanish patients with AR had significantly higher serum titers (*p* < 0.05) for six mite molecules (35.29%)—Der p 23, Der f 1, Der f 2, Gly d 2, Lep d 2, and Tyr p 2—compared to Peruvian subjects with AR. Additionally, they had significantly higher titers for 10 allergens (58.82%)—Der p 2, Der p 5, Der p 10, Der p 21, Der p 23, Der f 1, Der f 2, Blo t 10, Gly d 2, and Lep d 2—compared to the group of Peruvian asthmatics (Table 4).

#### 2.3.2. Age and sIgE Reactivity Profiles

Younger patients (<21 years old, *n* = 76) displayed a higher frequency of sIgE binding to 10 out of 17 molecular mite allergens—specifically, Der p 1, Der p 2, Der p 5, Der p 10, Der p 23, Der f 1, Blo t 5, Blo t 10, Gly d 2, and Tyr p 2—compared to older patients (≥21 years old, *n* = 105). Furthermore, younger subjects had a higher frequency of sIgE binding in both AR (14 molecules) and AA (9 molecules) compared to their older counterparts. Among adults, those with asthma showed a higher frequency of sIgE binding to 16 out of 17 mite molecules (excluding Tyr p 2) compared to adult patients with AR. Regarding geographical variations, younger patients with AR in the Spanish cohort demonstrated a higher frequency of sensitization to 11 mite molecules compared to their AA peers. Conversely, in the Peruvian cohort, younger AA subjects exhibited a higher frequency of sensitization to 11 out of 17 allergens compared to those with AR.

#### 2.3.3. IgE Western Blot

Western blot analysis of the allergenic patterns of *Dermatophagoides pteronyssinus* (DPT), *Blomia tropicalis* (BTR), and *Lepidoglyphus destructor* (LDE) were identified for each group of geographic patients. Notably, the ~14 and 16-kDa bands consistently were the most prominent in DPT, whereas, in BTR and LDE, it was a ~14 kDa protein band. The allergenic pattern of DPT and BTR was similar for the four study populations, while that of LDE showed a markedly higher recognition by the Tenerife West population. These protein bands could correspond to the most prevalent DPT, BTR, and LDE allergens in these populations, according to the molecular results cited above.

### 2.4. Cat and Dog Epithelia

Overall, 87 out of 181 patients (48.06%) were sensitized to one or more of the 10 investigated epithelial molecular allergens. Among these, cat allergens were the most frequently identified, with 67 patients (37.01%) showing sensitivity, compared to dog allergens in 45 patients (24.86%). A total of 112 specific IgE (sIgE) responses to cat allergens were detected, with Fel d 1 being the most common (33.57%), followed by Fel d 7 (6.7%), Fel d 4 (6.6%), and Fel d 2 (1.0%). For dog allergens, Can f 5 was the most frequently identified (19.1%), followed by Can f 1 (9.92%), Can f 4 (4.3%), Can f 6 (2.45%), Can f 2 (0.5%), and Can f 3 (0.5%).

Asthmatics, compared to AR patients, exhibited higher frequencies of Fel d 1, Fel d 4, Fel d 7, and Can f 1 in both the Spanish and Peruvian cohorts, while Can f 6 was observed only in the Spanish cohort. In contrast, Can f 5 was more frequently found in AR patients than in those with AA in both investigated populations. In the Spanish group, the most frequent epithelial allergens among AR patients were Can f 5 (28%) and Fel d 1 (24%), followed by Can f 4 and Can f 6 (6%), Can f 1 and Fel d 4 (4%), and Fel d 2, Fel d 7, Can f 2, and Can f 3 (2%). In AA patients, the frequencies were Fel d 1 (44.2%), Can f 5 (25%), Can f 1 (15.4%), Fel d 4 (13.5%), Fel d 7 (11.5%), Can f 4, and Can f 6 (5.8%), while Fel d 2, Can f 2, and Can f 3 were undetected. In the Peruvian cohort, the most frequent allergens for AR patients were Fel d 1 (32%), Can f 5 (18%), Can f 1 (10%), Fel d 4 and Fel d 7 (4%), and Fel d 2 and Can f 4 (2%), with Can f 2, Can f 3, and Can f 6 not detected. Among AA patients, the frequencies were Fel d 1 (34.5%), Can f 5 (13.8%), Fel d 7 and Can f 1 (10.3%), Fel d 4 (6.9%), and Can f 4 (3.4%), with no detection of Fel d 2, Can f 2, Can f 3, or Can f 6.

### 2.5. Pollens

A total of 37 out of 181 patients (20.44%) were sensitized to one or more of the 22 pollen allergens investigated, showing 58 sIgE individual pollen responses. The Spanish subset showed a higher detection of pollen molecules (23 molecules in 12 AR patients and 26 allergens in 13 AA subjects) compared to the Peruvian cohort (8 molecules in 6 AR patients and 1 molecule in 1 AA subject). 

In the Spanish cohort, the most frequent allergens for AR patients were Phl p 1 (10%), Lol p 1 (10%), Par j 2 (8%), Ole e 1 (4%), Art v 1 (4%), Cup a 1 (4%), Bet v 1 (2%), Art v 3 (2%), Pla a 3 (2%), Phl p 2 (2%), and Sal k 1 (2%). For AA patients, the frequencies were Pla a 3 (3.8%), Phl p 1 (9.6%), Lol p 1 (9.6%), Par j 2 (9.6%), Art v 1 (9.6%), Phl p 2 (3.8%), and Phl p 5 (1.9%). Similarly, in the Peruvian cohort, the most frequent allergens for AR patients were Par j 2 (8%), Ole e 7 (2%), Ole e 9 (2%), Phl p 2 (2%), and Phl p 5 (2%). For AA patients, Par j 2 was detected in 3.4% of cases.

### 2.6. Cockroach and Molds

Thirteen subjects out of 181 (7.18%) were sensitized to Bla g 9 and seven (3.86%) to Alt a 1. None of the patients showed sensitization to Alt a 6. In the Peruvian cohort, AR patients had sensitization rates of 10% for Bla g 9 and 2% for Alt a 1. For AA patients, Bla g 9 was detected in 3.4% of cases. In the Spanish cohort, sensitization rates for AR patients were 6% for Alt a 1 and 2% for Bla g 9, while for AA patients, the rates were 11.5% for Bla g 9 and 5.8% for Alt a 1. 

### 2.7. Regional Differences in IgE Profiles

Our investigation included 181 patients in total, with 90 individuals from Lima and 91 from Tenerife. Approximately 50% of participants from each location were from surrounding regional areas. Figure 2 presents the profiles of the 20 most frequently recognized molecular allergens in each region, highlighting the serodominance of each allergen. Significant differences (*p* < 0.05) were observed for allergens Der p 1, Der p 2, Der p 5, Der p 23, Der f 1, and Der f 2 between North and South Lima, Peru, and for Fel d 1 between East and West Tenerife, Spain.

## 3. Discussion

Climate change is influencing the seasonality, production, concentration, allergenicity, and geographic spread of airborne allergens, leading to increased rates of allergic diseases [25,26]. This study is the first to assess aeroallergen sensitization in two populations with respiratory allergies from distinct geographical and socio-economic backgrounds—Tenerife and Lima—both of which share similar climatic conditions.

Tenerife, the largest and highest island in the Canary archipelago, has a climate influenced by the cool, humid northeast trade winds from the Azores anticyclone. While generally having lower air pollution than continental Europe, Tenerife is occasionally impacted by Saharan dust intrusions, which can cause PM10 concentrations to exceed 50 µg/m^3^, a level linked to adverse health effects [27,28]. In contrast, Lima, Peru’s capital, is a densely populated city situated between the Pacific Ocean and the Andes Mountains. With over 11 million residents, Lima is the third-most populous and second-most polluted city in the Americas, affected by an aging transportation fleet, biomass stoves, and topography that traps pollutants [29,30]. Both Lima and Tenerife share a desert climate (Köppen BWh) but experience different seasonal patterns: Lima’s temperatures remain stable with minimal seasonal changes, while Tenerife, with its varied elevations, has more pronounced seasonal fluctuations [31,32].

Lima’s pollution levels, particularly PM2.5, CO_2_, and O_3_, are higher due to heavy traffic, industrial activity, and inefficient energy use [33,34,35]. Tenerife benefits from trade winds that disperse pollutants, though Saharan dust occasionally raises particulate levels. Additionally, Lima faces a significant income disparity (13:1) between its regions, leading to uneven pollution exposure, whereas Tenerife has a minimal income gap (1.1:1), reflecting more equitable socio-economic conditions (Figure 3) [36].

Specific IgE antibodies, detected in blood or via SPTs, are key indicators of allergic sensitization and strong predictors of respiratory symptoms in epidemiological studies [37,38]. In the current study, mites were consistently identified through both SPTs and serum sIgE as the most prevalent allergen affecting both populations. Among the population studied, 85% were sensitized to at least one of the 17 investigated mite molecular allergens, 48.06% to one or more of the 10 epithelial allergens, 20.44% to one or more of the 22 pollen allergens, 7.18% to Bla g 9, and 3.86% to Alt a 1. These findings highlight mites as a primary target for intervention while also revealing a diverse spectrum of other allergens that may contribute to the overall burden of respiratory symptoms.

The production of HDM allergens is intricately connected to various environmental factors, which could potentially amplify allergenicity in the context of climate change and pollution. House dust mites (HDMs) synthesize cysteine and serine proteases, including Der p 1, and these levels fluctuate based on dietary influences and temperature variations [39]. Additionally, lipid-binding proteins such as Der p 2 and Der p 7 play a crucial role in helping mites detect pathogens; their concentrations may rise in humid environments rich in fungal presence [40]. Pollution significantly impacts allergen production, as exposure to diesel particulate matter has been shown to enhance the synthesis of glutathione transferase allergens like Der p 8, indicating that deteriorating air quality is associated with increased allergenicity of HDMs [41,42]. Furthermore, this study reaffirms that Der p 1, Der p 2, Der f 2, and Der p 23 remain prominent allergens while simultaneously highlighting regional and disease-specific variations in sensitization patterns [43]. Notably, these differences are particularly pronounced among patients with allergic rhinitis (AR) compared to those with allergic asthma (AA), as well as between cohorts from Spain and Peru.

In patients with AR, the Spanish cohort showed significantly higher sensitization rates to key mite allergens, including Der p 23, Der f 1, and Lep d 2, compared to their Peruvian counterparts [22]. While some allergens, such as Der p 10, Blo t 10, Blo t 21, and Gly d 2, were commonly recognized in both groups, Spanish patients displayed higher overall IgE levels. These differences were even more pronounced in individuals with AA, where Spanish patients were more sensitized to 13 of the 17 allergens studied, including Der p 1, Der p 2, and Der p 23. In contrast, Peruvian AA patients showed higher sensitization to only one allergen, Blo t 21. Spanish patients also exhibited significantly elevated sensitization to allergens such as Der p 5, Blo t 10, and Gly d 2. These findings suggest significant regional differences in allergen exposure or immune responses. This is consistent with the work of Muddaluru and colleagues, who compared house dust mite sensitization profiles in allergic adults from Canada, Europe, South Africa, and the USA, finding that Spanish asthmatics are more sensitive to a broader range of allergens. Interestingly, younger patients tend to exhibit a more pronounced immune response—sIgE binding—to a wider variety of allergens, particularly among those with AR [44]. This observation implies that age, along with geographical factors such as climate, local allergen exposure, and genetic predispositions, may influence the development of specific sensitization profiles in different populations. These findings suggest significant regional differences in allergen exposure or immune responses, as also revealed by Muddaluru and coworkers [44] when comparing house dust mite sensitization profiles in allergic adults from Canada, Europe, South Africa, and the USA, with Spanish asthmatics being more sensitive to a broader range of allergens. Interestingly, younger patients tend to show a more pronounced immune response—sIgE binding—to a wider range of allergens, particularly among those with AR.

This observation implies that age, along with geographical factors such as climate, local allergen exposure, and genetic predispositions, may influence the development of specific sensitization profiles across different populations. Moreover, it necessitates a reevaluation of the prevalence-based classification proposed by Posa et al. [45] for *Dermatophagoides* allergens, suggesting that it may require nuanced refinement. Specifically, it would be advisable to incorporate Der p 5, Der p 7, and Der p 21 into Group A allergens, which includes molecules with a prevalence exceeding 40% for Tenerife, as well as Der p 5 and Der p 21 for Lima. This adjustment would complement the previously recognized allergens—Der p 1, Der p 2, and Der p 23—and enhance the robustness of the classification framework.

Regarding epithelial allergens [46,47], Spanish patients with allergic rhinitis (AR) exhibited significant sensitivities to Can f 5 (28%) and Fel d 1 (24%), indicating a notable prevalence of both cat and dog allergens. The Peruvian cohort also showed high sensitization rates to Fel d 1 (32%) and Can f 5 (18%). Among patients with allergic asthma (AA), sensitivities varied between the cohorts. In the Spanish cohort, Fel d 1 was particularly prevalent, with 44.2% of asthma patients demonstrating sensitivity. Other common cat allergens included Fel d 4 (13.5%) and Fel d 7 (11.5%). For dog allergens, Can f 1 was found in 15.4% of asthmatic patients, while Can f 6 was unique to this cohort. In contrast, the Peruvian cohort’s asthmatics showed high sensitization rates to Fel d 1 (34.5%), followed by Can f 5 (13.8%) and both Fel d 7 and Can f 1 (10.3%). Certain allergens, such as Can f 2, Can f 3, and Can f 6, were undetected in this population, emphasizing geographical differences in allergen exposure and sensitization patterns. These data align with the study by Ukleja-Sokołowska N. and colleagues, which found that Fel d 1 was the primary allergen for cat-sensitized individuals, affecting 93.9% of cat-allergic patients, while Can f 1 and Can f 5 were the most common allergens among dog-sensitized individuals [48].

Pollens, such as ragweed in North America, grasses in Europe, and Japanese cedar in Japan, have been shown to have prolonged or overlapping seasons, creating continuous allergenic exposure for sensitive individuals. Cross-reactivity between pollen types, such as birch and cypress in Europe, further complicates allergic responses and prolongs the duration of symptoms [49,50]. These findings suggest the need for region-specific allergen management and more accurate monitoring to improve outcomes for pollen-sensitized patients. Consequently, our study revealed significant differences between the Spanish and Peruvian cohorts. The Spanish cohort identified 23 pollen allergens among 12 patients with AR) and 26 allergens in 13 patients with allergic AA, whereas the Peruvian cohort detected only 8 allergens in 6 AR patients and only 1 allergen in 1 AA patient. The most frequently identified allergens among Spanish AR patients were Phl p 1 and Lol p 1 (both at 10%), while AA patients exhibited sensitivities to Pla a 3 (3.8%) and Phl p 1 (9.6%). In contrast, Peruvian AR patients were primarily sensitized to Par j 2 (8%), with other allergens such as Ole e 7 (2%) and Ole e 9 (2%) also present. These results suggest that the Spanish population is exposed to a greater variety of pollen allergens, leading to broader sensitization, while the Peruvian cohort shows a more limited range of sensitization. This contrasts with the data reported by Dramburg et al., which revealed a high prevalence of IgE responses to major allergens such as grass pollen (Phl p 1, Phl p 5, and Cyn d 1), olive (Ole e 1), and cypress (Cup a 1), along with variable responses to other airborne allergens and pan-allergens [51].

Finally, both populations exhibited sensitivities to Bla g 9 and Alt a 1; yet, the rates varied significantly, especially among allergic asthma (AA) patients in Spain, who demonstrated higher sensitization levels. Notably, the lack of sensitization to Alt a 6 in all patients indicates that this allergen may not be pertinent to these populations, underscoring the necessity for region-specific diagnostic and management strategies for allergic conditions. This study presents several limitations that may influence its findings and generalizability. First, the comparison between Spanish and Peruvian populations could introduce variability stemming from differing environmental exposures and genetic backgrounds. Additionally, the emphasis on a narrow range of allergens, primarily house dust mites, risks overlooking other significant contributors to sensitization. Furthermore, evaluating sensitization at a single time point does not account for temporal variations, and the absence of longitudinal data hampers a comprehensive understanding of allergic responses over time.

## 4. Materials and Methods

### 4.1. Subjects

Between December 2023 and May 2024, we consecutively recruited children and adults aged 5 to 75 years with an allergist-confirmed diagnosis of allergic rhinitis (AR) and/or asthma (A) from the Outpatient Allergy Clinic and Severe Asthma Unit at Hospital Universitario de Canarias in Tenerife, Spain, as well as from Lima, Peru. Patients were recruited from two distinct areas within each region: East and West Tenerife in Spain and North and Central Lima in Peru. This investigation was reviewed and approved by the local Ethical Committee (code number CHUC_2023_66), and informed consent was obtained from all participants (and from parents/guardians for participants under 18 years old) upon their inclusion in this study. Patients were also required to have had clinical symptoms for at least three years after establishing local residency to meet the inclusion criteria. The severity and stage of allergic diseases were clinically evaluated following specific guidelines [25,26].

Clinical data collected from patients’ medical records included sociodemographic information, clinical history (including past medical conditions and current allergy diagnosis), and details about their medications. In line with routine clinical practice, only patients with a positive SPT to relevant aeroallergen extracts (such as mites, pollens, molds, or animal epithelia) were included in this study. Patients who had undergone past or current allergen immunotherapy or treatment with monoclonal antibodies (biologics), as well as pregnant and breastfeeding women, were excluded.

### 4.2. Skin Prick Test

Percutaneous testing was conducted according to European standards using a diagnostic panel (Inmunotek, Madrid, Spain) with standardized extracts, including *Dermatophagoides pteronyssinus* (*D*. *pteronyssinus*), *Blomia tropicalis* (*B*. *tropicalis*), *Lepidoglyphus destructor* (*L*. *destructor*), cat and dog dander, a grass mix (*Poa pratensis*, *Dactylis glomerata*, *Lolium perenne*, *Phleum pratense*, and *Festuca pratensis*), *Olea europaea*, *Betula verrucosa*, *Quercus ilex*, *Platanus hispanica*, *Plantago lanceolata*, *Parietaria judaica*, *Salsola kali*, *Artemisia vulgaris*, *Juniperus oxycedrus*, *Cupressus arizonica*, *Cupressus sempervirens*, and *Alternaria alternata* [52]. Histamine (10 mg/mL) and saline were used as positive and negative controls, respectively. As per standard procedure, antihistamines were discontinued one week before the SPT. Wheal diameters were measured 20 min after testing; diameters greater than 3 mm were considered positive.

### 4.3. Serological Analysis

Blood samples were collected from all participants, labeled with a unique code, stored at −40 °C, and thawed immediately before in vitro testing. Total IgE and specific IgE (sIgE) levels were measured using the ALEX MacroArray platform (MacroArray Diagnostics, Vienna, Austria) according to the manufacturer’s protocol. ALEX is a multiplex array containing 282 reagents, including 157 whole allergens and 125 molecular components. The allergens are attached to polystyrene nanobeads, which are then deposited on a nitrocellulose membrane, as previously described [53]. The assay included 17 mite molecular allergens: Der p 1; Der p 2; Der p 5; Der p 7; Der p 10; Der p 11; Der p 20; Der p 21; Der p 23; Der f 1; Der f 2; Blo t 5; Blo t 10; Blo t 21; Lep d 2; Gly d 2; and Tyr p 2. Additionally, 10 cat and dog epithelial allergens were tested: Fel d 1; Fel d 2; Fel d 4; Fel d 7; Can f 1; Can f 2; Can f 3; Can f 4; Can f 5; and Can f 6. A total of 22 pollen allergens were included: Bet v 1; Bet v 2; Bet v 6; Cup a 1; Pla a 1; Pla a 2; Pla a 3; Ole e 1; Ole e 7; Ole e 9; Phl p 1; Phl p 2; Phl p 5; Phl p 6; Phl p 7; Phl p 12; Lol p 1; Sal k 1; Pla l 1; Par j 2; Art v 1; and Art v 3. To evaluate mold sensitization, Alt a 1 and Alt a 6 were assessed, while Bla g 9 was tested for cockroach sensitization [54]. Total IgE levels were reported in international units per milliliter (IU/mL), while sIgE levels were expressed in kU_A_/L, with values ≥ 0.35 kU_A_/L considered positive.

Considering the specific IgE results, dust mite-positive patients were selected to prepare four serum pools. The serum pools were prepared using an equal amount of these individual patient sera from each region (North and South of Lima and East and West of Tenerife). The optimal dilution of each serum pool was obtained by ELISA assay (approximately 1.5 OD).

### 4.4. ELISA Assay

For the ELISA assay, Microlon high-binding plate wells (Greiner Bio-One, Frickenhausen, Germany) were coated with 1 µg of DPT, BTR, and LDE extracts in sodium carbonate/bicarbonate buffer at 4 °C overnight (ON). After washing with phosphate-buffered saline (PBS), 0.25% Tween 20 (PBS-T) wells were blocked with PBS-T, 1% bovine serum albumin (BSA) for 3 h at room temperature (RT). After washing, 100 µL of different dilutions of serum pools in PBS-T-BSA were added and then incubated ON at RT. The bound IgE antibodies were detected by incubation with 100 µL of Horseradish Peroxidase (HRP)-conjugate anti-human IgE (Southern Biotech B3102E8). The substrate o-phenylenediamine dihydrochloride (Sigma 34006) diluted in 10 mL of phosphate–citrate buffer 0.1 M, 0.025% peroxide hydrogen was added for the development of reactivity, and OD was measured in an ELISA reader at 405 nm.

### 4.5. IgE Western Blot

Proteins from *D. pteronyssinus, B. tropicalis*, and *L. destructor* extracts were separated by Any kD™ Mini-PROTEAN^®^ TGX™ precast polyacrylamide gels (Bio-Rad Laboratories, Hercules, CA, USA) under reducing conditions according to the Laemmli’s method [55]. Proteins were electro-transferred to 0.45 µm nitrocellulose membranes, and the binding of IgE antibody to allergens was analyzed using pool serum of each region and anti-human IgE peroxidase conjugate (Southern Biotech, Biotechnology Research, Birmingham, AL, USA). Chemiluminescence detection reagents (Western Lightning PlusECL, Perkin Elmer, Walthman, MA, USA) were added following the manufacturer´s instructions, and the image was analyzed in Image Lab Touch software 3.0.1.14. IgE-binding bands were identified using the BioRad Diversity database program (Bio-Rad Laboratories, Hercules, CA, USA).

### 4.6. Statistical Analysis

Demographic characteristics were summarized using medians and standard deviations for continuous variables and percentages for categorical variables. To compare differences, analysis of variance (ANOVA) was used for parametric continuous variables, the Kruskal–Wallis and Mann–Whitney U tests for nonparametric continuous variables, and the Chi-square test for categorical variables. A *p*-value of less than 0.05 was considered statistically significant. All statistical analyses were performed using GraphPad Prism version 10.0.0 for Windows (GraphPad Software, La Jolla, CA, USA).

## 5. Conclusions

In summary, this study highlights significant geographical and disease-specific variations in allergen sensitization between Spanish and Peruvian patients. Notably, individuals in Spain, particularly those with asthma, showed higher sensitization levels to a broader range of allergens, suggesting differences in exposure or immune responses to house dust mites. These insights are critical for developing region-specific diagnostic and treatment strategies for allergic diseases. Additionally, the findings underscore the prominent role of mites as primary allergens in both populations, revealing distinct sensitization patterns influenced by local environmental and genetic factors. As climate change continues to impact airborne allergens—altering their allergenicity, seasonality, production, and atmospheric concentration—identifying local external exposures becomes increasingly vital. This research lays the groundwork for personalized diagnostic and therapeutic approaches tailored to the unique clinical phenotypes of allergies, ultimately improving the effectiveness of interventions aimed at reducing the burden of allergic diseases across diverse populations.

## Figures and Tables

**Figure 1 ijms-26-00284-f001:**
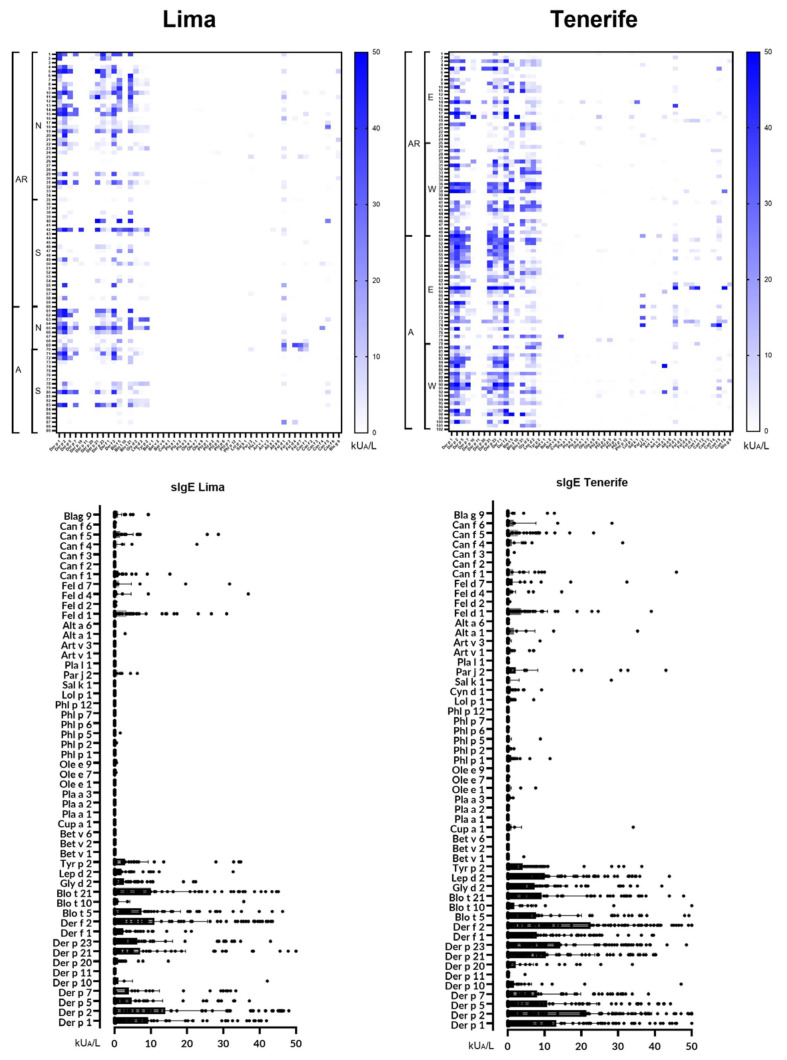
Heatmap illustrating the distribution of airborne allergens in Lima (Peru) and Tenerife (Spain), accompanied by their serodominance levels. Statistically significant differences (*p* > 0.05) were observed between Lima and Tenerife for the following allergens: Der p 2; Der p 5; Der p 10; Der p 21; Der p 23; Der f 2; Lol p 1; Pla l 1; Art v 3; and Alt a 6. AR, allergic rhinitis; A, asthma; N, north; S, south; E, east; W, west.

**Figure 2 ijms-26-00284-f002:**
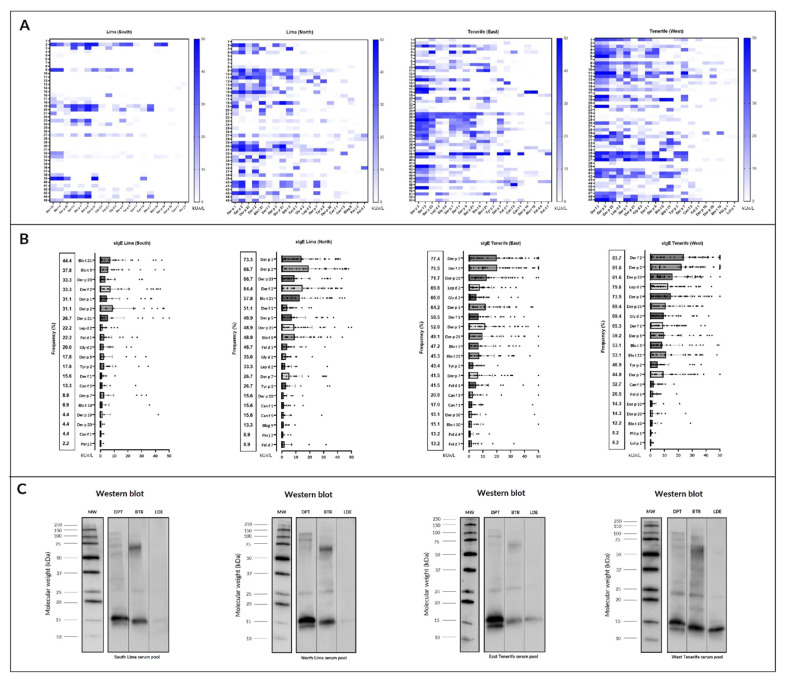
Sensitization profile to specific IgE (sIgE): (**A**) heatmap and (**B**) scatter plots with bars of the 20 most frequently identified molecular allergens in each region, along with their serodominance. Significant statistical differences (*p* < 0.05) were found for Der p 1, Der p 2, Der p 5, Der p 23, Der f 1, and Der f 2 between the two zones in Lima and for Fel d 1 between the two zones in Tenerife. (**C**) IgE Western blot of the different geographic groups included Dermatophagoides pteronyssinus (DPT), Blomia tropicalis (BTR), and Lepidoglyphus destructor (LDE). Different patterns of sIgE-binding were identified for each group of patients.

**Figure 3 ijms-26-00284-f003:**
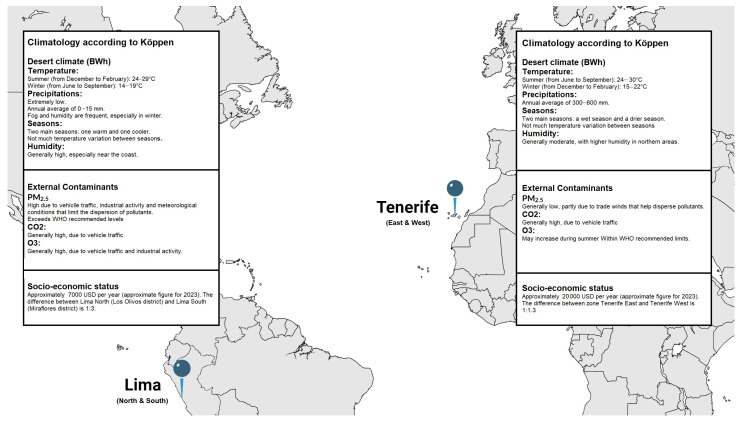
Climatology, external contaminants, and socio-economic status between North and South Lima, Peru, and East and West Tenerife, Spain.

**Table 1 ijms-26-00284-t001:** Descriptive statistics regarding basal comorbid conditions and associated clinical characteristics of the studied population (*n* = 181).

	Tenerife, Spain	Lima, Peru
*n* = 181	91	90
Age (y.o.) median (range)	29.75 (8–70)	22.0 (4–75)
<21 y.o. (*n* = 76)	38 (41.7%)	38 (42.2%)
≥21 y.o. (*n* = 105)	53 (58.2%)	52 (57.7%)
Sex (F/M)	61/30	66/24
Allergic Rhinitis (*n* = 135)	46 (50.5%)	89 (98.8%)
Allergic Asthma (*n* = 75)	45 (49.5%)	30 (33.3%)
SPT + to any aeroallergen	91 (100%)	90 (100%)
Total IgE (IU/mL) median (range)	365 (23.88–1010)	434.84 (46.32–762.3)

**Table 2 ijms-26-00284-t002:** Prevalence of sensitization to grouped local aeroallergens by Skin Prick Test (SPT) (*n* = 181). Bold figures represent significant differences (*p* < 0.05) in SPT-determined sensitization to grouped allergens between the two investigated populations.

Positive SPT (*n* = 181)	Tenerife, Spain (*n* = 91)	Lima, Peru (*n* = 90)
HDM and/or SM (%)	**88 (96.7)**	68 (75.5)
Cat and/or dog dander (%)	**34 (37.3)**	14 (15.6)
Pollen (%)	12 (13.1)	14 (15.8)
Cockroach (%)	7 (7.6)	10 (11.1)
Molds (%)	6 (6.5)	2 (2.2)

SPT: Skin Prick Test. HDM: House Dust Mites. SM: Storage Mites.

**Table 3 ijms-26-00284-t003:** Summary of identified mite molecular allergens (any), geographical locations, and associated baseline atopic diseases—allergic rhinitis (AR) and/or allergic asthma (AA)—in 181 patients analyzed using a microarray.

Number of Identified Mite Molecular Allergens (Any)	AR Tenerife, Spain	AR Lima, Peru	AA Tenerife, Spain	AA Lima, Peru
0	0	18	3	6
1	1	1	1	2
2	0	3	1	1
3	2	2	1	2
4	5	7	1	2
5	3	3	0	1
6	5	5	4	2
7	6	3	1	2
8	3	2	4	2
9	7	3	5	3
10	4	5	3	0
11	6	3	11	1
12	5	2	5	1
13	1	1	7	2
14	1	2	2	1
15	1	0	0	0
16	0	0	1	0
17	0	0	0	0

**Table 4 ijms-26-00284-t004:** Serological analysis presenting the mean (and frequency) of specific IgE (sIgE) concentrations (kU/L) against mite molecular allergens in patients with allergic rhinitis (AR, *n* = 135) and allergic asthma (AA, *n* = 75). Bold values denote statistically significant differences (*p* < 0.05) in mean sIgE levels between the two atopic conditions.

Molecule	sIgE AR (Spain)	sIgE AR (Peru)	sIgE AA (Spain)	sIgE AA (Peru)
**Der p 1**	11.7 ± 15.6 (66)	5.52 ± 10 (46)	13.6 ± 13.4 (82.7)	10.5 ± 14.4 (51.7)
**Der p 2**	16.19 ± 17.3 (76)	10.7 ± 16.7	**25.23** **±** **18.3**	16 ± 19.7
**Der p 5**	8.14 ± 14.1(42)	3.36 ± 8.5 (28)	**12.15** **±** **13.6 (69.2)**	4.29 ± 8.3 (34.5)
**Der p 7**	5.08 ± 10.3 (30)	2.54 ± 7.6 (16)	9.62 ± 13.7 (55.8)	5.31 ± 10.7 (24.1)
**Der p 10**	1.67 ± 7.2 (14)	0.9 ± 5.9 (4)	**0.65** **±** **2.2 (15.4)**	0 ± 0 (0)
**Der p 11**	0 ± 0 (0)	0 ± 0 (0)	0 ± 0 (1.9)	0 ± 0 (0)
**Der p 20**	1.72 ± 6 (10)	0.35 ± 2.1 (6)	1.47 ± 4.7 (15.4)	0.55 ± 1.6 (13.8)
**Der p 21**	7.38 ± 12.45 (48)	5.61 ± 12.9 (32)	**12.16** **±** **14.8 (69.2)**	6.58 ± 12.1 (41.4)
**Der p 23**	**13.44** **±** **14.8 (72)**	3.25 ± 6.6 (42)	**13.93** **±** **14.5 (80.8)**	6.06 ± 10.5 (51.7)
**Der f 1**	**3.63** **±** **5.5 (50)**	0.97 ± 3 (22)	**10.96** **±** **11.9 (73.1)**	2.38 ± 4.7 (44.8)
**Der f 2**	**17.17** **±** **18.3 (76)**	8.15 ± 13.5 (44)	**26.5** **±** **18.7 (82.7)**	12.54 ± 16.8 (51.7)
**Blo t 5**	7.85 ± 12.9 (46)	5.71 ± 10.5 (36)	6.39 ± 12.9 (53.8)	4.43 ± 6.8 (44.8)
**Blo t 10**	1.78 ± 8.1 (12)	0.81 ± 5 (8)	**0.61** **±** **1.9 (15.4)**	0 ± 0 (0)
**Blo t 21**	10.46 ± 13.7 (54)	7.35 ± 1 (46)	6.99 ± 11.9 (44.2)	8.19 ± 12.8 (51.7)
**Gly d 2**	**7.34** **±** **10.2 (74)**	1.31 ± 3.4 (22)	**6.11** **±** **8.8 (61.5)**	1.81 ± 3.2 (27.6)
**Lep d 2**	**9.92** **±** **12 (72)**	0.67 ± 1.67 (22)	**9.12** **±** **10.1 (76.9)**	2.34 ± 6.5 (27.6)
**Tyr p 2**	**4.41** **±** **8.6 (46)**	1.54 ± 5.3 (20)	2.43 ± 4.2 (44.2)	3.1 ± 8.8 (17.2)

## Data Availability

The data that support the findings of this study are available from Servicio Canario de Salud but restrictions apply to the availability of these data, which were used under license for the current study and so are not publicly available. Data are, however, available from the authors upon reasonable request and with permission of Servicio Canario de Salud.

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
