# Peer review of "Molecular Signatures of Aeroallergen Sensitization in Respiratory Allergy: A Comparative Study Across Climate-Matched Populations"

_ijms, 2024, doi:10.3390/ijms26010284_

Round 1

Reviewer 1 Report

Comments and Suggestions for Authors

This is a very interesting and pioneer study comparing sensitization to airborne allergens in 2 distant yet climate-matched populations of the Canaries and Peru. Please find below my comments:

Introduction: Provides a detailed and thorough account of the rationale for the study and an overview of the environmental and climate influence on allergic sensitization prevalence. Nothing to add on my part.

Discussion addresses all major points in findings and provides complete explanation and significance of the results obtained by the Authors. Nothing to add on my part, either.

Minor and editorial issues to be addressed:

Table 2: have there been any statistical differences in SPT-assessed sensitization between the populations with regard to each allergen?

Figure 1: should be enlarged to make the information more readable

Figure 2C: the explanation of the Western Blot images seems to be incomplete. Which populations/groups are represented in each image? Which molecules are represented on the molecular weight ladder? Please explain and/or amend.

Line 78: I suggest rephrasing to: “… drug allergies and/or hypersensitivities”  _ NSAOIDs may cause non-immunologically dependent symptoms as well.

Lines 82-83; I suggest rephrasing to: “…and its prevalence in 181 patients…”

The fragments in lines 283 through 286 and 292 through 296 seem almost identical and information is repeated. Please check again if it was intended or due to mistake.

Latin names of species should be in italics – throughout the manuscript.

Author Response

Answers to Reviewer #1

This is a very interesting and pioneer study comparing sensitization to airborne allergens in 2 distant yet climate-matched populations of the Canaries and Peru. Please find below my comments:

Introduction: Provides a detailed and thorough account of the rationale for the study and an overview of the environmental and climate influence on allergic sensitization prevalence. Nothing to add on my part.

Discussion addresses all major points in findings and provides complete explanation and significance of the results obtained by the Authors. Nothing to add on my part, either.

Minor and editorial issues to be addressed:

Table 2: have there been any statistical differences in SPT-assessed sensitization between the populations with regard to each allergen?

Thank you for the accurate comment. Statistical differences among allergens and both populations have been addressed in the new version of Table 2.

From:

Table 2. Prevalence of sensitization to grouped local aeroallergens by Skin Prick Test (n= 181).

Positive SPT (n=181)

Tenerife, Spain (n=91)

Lima, Peru (n=90)

HDM and/or SM (%)

88 (96.7)

68 (75.5)

Cat and/or dog dander (%)

34 (37.3)

14 (15.6)

Pollen (%)

12 (13.1)

14 (15.8)

Cockroach (%)

7 (7.6)

10 (11.1)

Molds (%)

6 (6.5)

2 (2.2)

To:

Table 2. Prevalence of sensitization to grouped local aeroallergens by Skin Prick Test (SPT) (n= 181). Bold figures represent significant differences (P < 0.05) in SPT-determined sensitization to grouped allergens between the two investigated populations.

Positive SPT (n=181)

Tenerife, Spain (n=91)

Lima, Peru (n=90)

HDM and/or SM (%)

88 (96.7)

68 (75.5)

Cat and/or dog dander (%)

34 (37.3)

14 (15.6)

Pollen (%)

12 (13.1)

14 (15.8)

Cockroach (%)

7 (7.6)

10 (11.1)

Molds (%)

6 (6.5)

2 (2.2)

Figure 1: should be enlarged to make the information more readable.

Thanks for the consideration. We have enlarged the picture to make the information more readable.

Figure 2C: the explanation of the Western Blot images seems to be incomplete. Which populations/groups are represented in each image? Which molecules are represented on the molecular weight ladder? Please explain and/or amend.

Thanks for the consideration. We have detailed the explanation about which populations/groups are represented in each image, and regarding the molecular weight ladder, we have carried out with a commercial standard (Precision Plus Protein™ WesternC™ Blotting standard)

Line 78: I suggest rephrasing to: “… drug allergies and/or hypersensitivities” _ NSAOIDs may cause non-immunologically dependent symptoms as well.

Thanks for the consideration. We have just replaced the phrase.

Lines 82-83; I suggest rephrasing to: “…and its prevalence in 181 patients…”

Thanks for the consideration. We have just replaced this phrase.

The fragments in lines 283 through 286 and 292 through 296 seem almost identical and information is repeated. Please check again if it was intended or due to mistake.

Thanks for the consideration. We have just modified this paragraph:

In patients with AR, the Spanish cohort showed significantly higher sensitization rates to key mite allergens, including Der p 23, Der f 1, and Lep d 2, compared to their Peruvian counterparts [22]. While some allergens, such as Der p 10, Blo t 10, Blo t 21, and Gly d 2, were commonly recognized in both groups, Spanish patients displayed higher overall IgE levels. These differences were even more pronounced in individuals with AA, where Spanish patients were more sensitized to 13 of the 17 allergens studied, including Der p 1, Der p 2, and Der p 23. In contrast, Peruvian AA patients showed higher sensitization to only one allergen, Blo t 21. Spanish patients also exhibited significantly elevated sensitization to allergens such as Der p 5, Blo t 10, and Gly d 2.

Latin names of species should be in italics – throughout the manuscript.

Corrected. Thank you.

Reviewer 2 Report

Comments and Suggestions for Authors

Molecular Signatures of Aeroallergen Sensitization in Respiratory Allergy: A Comparative Study Across Climate Matched Populations

This study explores aeroallergen sensitization in populations from Tenerife, Spain, and Lima, Peru, using the Skin Prick Test and serological IgE analysis. However, the manuscript's organization and the interpretation of the results are unsatisfactory.

Major comments

1.Focus on highlighting and interpreting the findings directly within the results section rather than deferring these discussions to the discussion section. The manuscript's flow is currently problematic, making it challenging to follow and comprehend. Specifically, the explanations related to Table 3 and Table 4 are particularly difficult to understand. For instance, in lines 118-121, Table 3 is referenced to explain details including Der p 23, which is notably absent from Table 3.

2.The authors should provide a detailed explanation of how the serum pools from each region were prepared for the western blot analysis. This should include specific information about the collection, processing, and pooling methods to ensure clarity and reproducibility.

3.Many of the results presented in the manuscript lack proper interpretation. For instance, the western blot results are reported without providing a clear explanation or analysis of their significance. The authors should include detailed interpretations to enhance the understanding and impact of the findings.

Author Response

Answers to Reviewer #2.

This study explores aeroallergen sensitization in populations from Tenerife, Spain, and Lima, Peru, using the Skin Prick Test and serological IgE analysis. However, the manuscript's organization and the interpretation of the results are unsatisfactory.

Major comments

1.Focus on highlighting and interpreting the findings directly within the results section rather than deferring these discussions to the discussion section. The manuscript's flow is currently problematic, making it challenging to follow and comprehend. Specifically, the explanations related to Table 3 and Table 4 are particularly difficult to understand. For instance, in lines 118-121, Table 3 is referenced to explain details including Der p 23, which is notably absent from Table 3.

Thank you for the precise comments. The following changes have been made to improve clarity and readability.

From: Table 3: Number of identified mite molecular allergens, geographical location, and corresponding basal atopic disease (allergic rhinitis (AR), and alergic asthma (AA)) in 181 patients studied with microarray.

To:  Table 3: Summary of identified mite molecular allergens (any), geographical locations, and associated baseline atopic diseases -allergic rhinitis (AR) and/or allergic asthma AA)- in 181 patients analyzed using a microarray.

From: Table 4: Serological analysis - Mean (frequency) of specific IgE (sIgE) responses (kU/L) to mite molecular allergens in patients with allergic rhinitis (AR, n=135) and allergic asthma (AA, n=75). Bold figures indicate significant differences (P < 0.05) in mean sIgE levels to mite molecular allergens between the two atopic conditions.

To: Table 4: Serological analysis presenting the mean (and frequency) of specific IgE (sIgE) concentrations (kU/L) against mite molecular allergens in patients with allergic rhinitis (AR, n = 135) and allergic asthma (AA, n = 75). Bold values denote statistically significant differences (P < 0.05) in mean sIgE levels between the two atopic conditions.

From: Lines118-121: Most subjects (57%) recognized between 9 and 16 of the tested allergens. The most common IgE profile, comprising 11 molecules, included all five major allergens. Single sensitizations to mite allergens were rare (2.76%), with only 1 out of 181 patients (0.5%) showing exclusive sensitization to Der p 23 (Table 3).

To: A majority of subjects (57%) demonstrated recognition of 9 to 16 individual mite allergens (any), with the predominant IgE profile—consisting of 11 molecules—covering all five major allergens (Table 3). Isolated sensitizations to individual mite allergens were infrequent, occurring in only 2.76% of cases, with exclusive sensitization to Der p 23 observed in a single patient (0.5%) out of 181 (data not shown).

2.The authors should provide a detailed explanation of how the serum pools from each region were prepared for the western blot analysis. This should include specific information about the collection, processing, and pooling methods to ensure clarity and reproducibility.

Thanks for the consideration. According with that, we have included the following specific information on how we prepared serum pools:

Considering the specific IgE results, dust mite-positive patients were selected to prepare four serum pools. The serum pools were prepared using an equal amount of these individual patient sera from each region (North and South of Lima, and East and West of Tenerife). The optimal dilution of each serum pool was obtained by ELISA assay (approximately 1.5 of OD).

4.4. ELISA assay

For the ELISA assay, Microlon high-binding plate wells (Greiner Bio-One, Frickenhausen, Germany) were coated with 1 µg of DPT, BTR and LDE extracts in sodium carbonate/bicarbonate buffer at 4 â—¦C overnight (ON). After washing with phosphate buffered saline (PBS), 0.25% Tween 20 (PBS-T) wells were blocked with PBS-T, 1% bovine serum albumin (BSA) for 3 h at room temperature (RT). After washing, 100 µL of different dilutions of serum pools in PBS-T-BSA were added and then incubated ON at RT. The bound IgE antibodies were detected by incubation with 100 µL of Horseradish Peroxidase (HRP)-conjugate anti human IgE (Southern Biotech B3102E8). The substrate o-phenylenediamine dihydrochloride (Sigma 34006) diluted in 10 mL of phosphate-citrate buffer 0.1 M, 0,025% peroxide hydrogen was added for the development of reactivity, and OD was measured in an ELISA reader at 405 nm.

3.Many of the results presented in the manuscript lack proper interpretation. For instance, the western blot results are reported without providing a clear explanation or analysis of their significance. The authors should include detailed interpretations to enhance the understanding and impact of the findings.

Thanks for the consideration. We have just added one paragraph (2.3.3) in the results part, where we have included interpretations of western blot results.

2.3.3. IgE Western blot

Western blot analysis of the allergenic patterns of Dermatophagoides pteronyssinus (DPT), Blomia tropicalis (BTR) and Lepidoglyphus destructor (LDE) were identified for each group of geographic patients. Notably, the ~14 and 16-kDa bands consistently were the most prominent in DPT. Whereas in BTR and LDE it was a ~14 kDa protein band. The allergenic pattern of DPT and BTR was similar for the four study populations, while that of LDE showed a markedly higher recognition by the Tenerife West population. These protein bands could correspond to the most prevalent DPT, BTR and LDE allergens in these populations, according to the molecular results cited above.

Round 2

Reviewer 2 Report

Comments and Suggestions for Authors

The authors have responded to the revision. This reviewer is satisfied with the modifications in the manuscript and believes that the manuscript now has been improved for publication.